# Plan-and-Paint: Collaborating Semantic and Noise Reasoning for Text-to-Image Generation

## Abstract

Despite the transformative success of chain-of-thought (CoT) and reinforcement learning (RL) in large language models, their application to visual generation—where reasoning is a critical challenge—remains largely unexplored. In this paper, we present **Plan-and-Paint**, a novel framework that integrates a dual-level reasoning hierarchy for text-to-image generation. Our framework operates at two critical stages: (1) at the semantic level, an adaptive planner first decomposes the input prompt into a structured generation plan, and (2) at the foundational level, a reinforcement learning agent optimizes the initial noise prior to align with this plan. To seamlessly coordinate these two stages, we introduce a unified reinforcement learning paradigm GRPO to jointly optimizes both the planning coherence and the execution fidelity through a composite reward function. Extensive experiments demonstrate the superiority of our approach: Plan-and-Paint achieves significant improvements on both GenEval (0.87→0.90) and WISE benchmarks. Most importantly, on GenEval benchmark, our method secures the top rank, outperforming a wide range of top-tier open-source and closed-source competitors, including GPT-Image-1 High, Janus-Pro-7B, Qwen-Image, BAGEL, and Seedream 3.0 by a significant margin. Our work advances the state-of-the-art in text-to-image generation, proving that an explicit reasoning hierarchy is key to unlocking controllable and compositional text-to-image generation. To facilitate future research, we will make our code and pre-trained models publicly available.

## 1 Introduction

Visual generation, particularly through diffusion models (Saharia et al., 2022; Podell et al., 2023; Wang et al., 2025), has achieved remarkable success in synthesizing high-fidelity images from natural language descriptions. Despite their impressive performance, these models remain constrained by their reliance on purely random initial noise that is entirely agnostic to the target semantic content. This semantic-agnostic initialization necessitates computationally intensive blind exploration through multiple denoising steps before meaningful structures begin to emerge (Ho et al., 2020). This limitation is further exacerbated in fast-sampling techniques like mean flow (Geng et al., 2025), where the deterministic generation trajectory makes outputs critically dependent on the initial noise condition. NoiseAR (Li et al., 2025) tackles this by introducing an autoregressive model that learns a conditional and semantically rich noise prior. However, NoiseAR's reasoning capability is acquired through supervised training on annotated data of text-noise pairs, which inherently limits its ability to generalize to novel compositional concepts or dynamically reason about unseen configurations, making it incapable of semantic textual reasoning. For instance, when given an ambiguous prompt like "Traditional food of the Mid-Autumn Festival", NoiseAR cannot infer the intended concept (e.g., "mooncake") and often generates unfaithful results due to its reliance on superficial textual correlations from training data, as shown in Fig. 1.

Recent advances in large language models (LLMs), such as OpenAI o1 (OpenAI, 2024) and DeepSeek-R1 (Guo et al., 2025), have demonstrated significant capabilities in complex reasoning across domains, including mathematics (Amini et al., 2019; Hendrycks et al., 2021; Shao et al., 2024), coding (Chen et al., 2021; Austin et al., 2021; Jain et al., 2024), and writing (Cardon et al., 2023; Achiam et al., 2023). By incorporating reinforcement learning (RL) techniques, these mod-

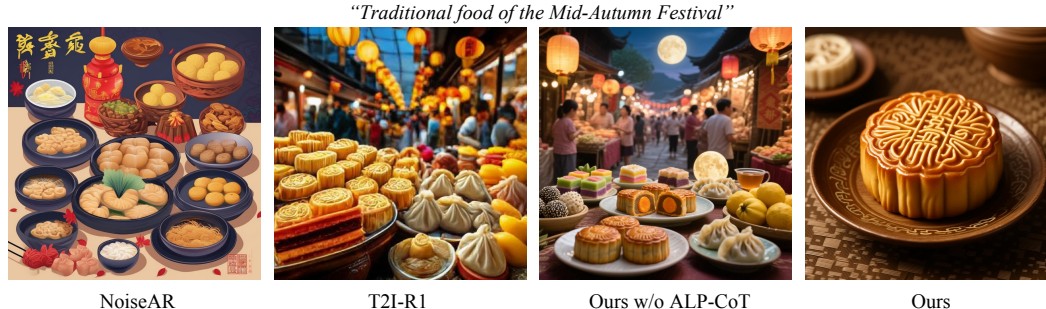

*"Traditional food of the Mid-Autumn Festival"*

NoiseAR       T2I-R1       Ours w/o ALP-CoT       Ours

Figure 1: Given the prompt *"Traditional food of the Mid-Autumn Festival"*, NoiseAR (Li et al., 2025) fails to generate a discernible mooncake. T2I-R1 (Jiang et al., 2025) produces a blurry image with a cluttered background. Ours w/o ALP-CoT, which only uses fixed-length semantic-level reasoning, suffers from over-depiction, introducing irrelevant objects despite a clear mooncake. Our method generates a high-quality image that accurately highlights the mooncake, precisely focusing on the theme.

els utilize structured Chain-of-Thought (CoT) (Wei et al., 2022) reasoning to decompose problems into sequential steps, substantially improving inference reliability. Inspired by these developments, generative vision models have begun integrating CoT-style mechanisms to enhance semantic coherence, particularly in autoregressive image synthesis. Recent work in text-to-image generation has increasingly adopted prompt rewriting (Deng et al., 2025) and semantic-level planning (Jiang et al., 2025; Duan et al., 2025) before image generation. Methods like T2I-R1 (Jiang et al., 2025) employ semantic-level CoT reasoning, where input text is reinterpreted into structured and detailed generative descriptions to better guide the image generation process.

However, we observe that naively applying a fixed semantic-level prompt CoT is fundamentally suboptimal. As illustrated in Fig. 1, excessive or indiscriminate elaboration can dilute primary subject information, introduce extraneous contextual details, and ultimately compromise both semantic alignment and image quality. To address this challenge, we propose an Adaptive Length Prediction for CoT (ALP-CoT) mechanism. In contrast to existing fixed-length CoT methods, our approach dynamically modulates the elaboration extent through an explicit assessment of input ambiguity, object-relation complexity, and attribute-binding specificity. By expanding prompts into structured descriptive chains only when necessary, our method ensures a balance between conciseness and expressiveness, thereby substantially improving the fidelity and relevance of generated outputs.

Despite recognizing the importance of semantic-level prompt CoT, a critical challenge remains unaddressed: How to effectively leverage and optimize such reasoning processes within a reinforcement learning framework tailored for visual generation? Extending reinforcement learning to visual generation introduces complexities distinct from those in code, mathematics, or conventional language tasks. Designing reward functions that capture the multidimensional nature of image quality—including semantic fidelity, spatial accuracy, attribute binding, overall coherence, and aesthetic appeal—poses significant difficulties. Therefore, effective reinforcement learning for visual generation necessitates a comprehensive reward framework that evaluates generated images from multiple dimensions to ensure reliable quality assessment, while also functioning as a regularization method to prevent it hacking a single reward model.

To address these challenges, we propose a multi-reward framework that integrates specialized vision-language experts to provide robust evaluation. We incorporate a human preference model (Wu et al., 2023) for aesthetic and semantic alignment, an open-vocabulary detector (Liu et al., 2024) for object existence and spatial relations, and a Visual Question Answering (VQA) model (Wang et al., 2022) for fine-grained attribute binding and theme clarity. This design ensures comprehensive supervision across aesthetic, structural, semantic, and theme dimensions while preventing over-optimization to individual rewards. Combined with Group Relative Policy Optimization (GRPO), our approach enhances reasoning and generalization to complex prompts. Experiments on GenEval (Ghosh et al., 2023) demonstrate that our method not only achieves state-of-the-art performance but also secures the top rank, significantly outperforming open-source and closed-

source strong competitors including GPT-Image-1 High (OpenAI, 2025), Janus-Pro-7B (Chen et al., 2025), Qwen-Image (Wu et al., 2025a), BAGEL (Deng et al., 2025), and Seedream 3.0 (Gao et al., 2025), demonstrating our method's effectiveness in achieving faithful and controllable text-to-image generation.

In summary, our main contributions are:

- We propose Plan-and-Paint, a novel dual-level reasoning framework that synergizes a high-level adaptive-length semantic planner and a low-level controlled noise-space executor, mirroring a human-like "plan-and-execute" creative paradigm.
- We design an adaptive-length prediction CoT that dynamically adjusts how much a prompt is elaborated, which improves accuracy and reduces errors from unnecessary details.
- We develop a multi-reward reinforcement learning framework integrating vision-language experts for comprehensive evaluation across aesthetic, structural, semantic, and theme dimensions, effectively preventing reward hacking.
- We achieve state-of-the-art performance on the GenEval benchmark, surpassing strong baselines like GPT-Image-1 High, Janus-Pro-7B, Qwen-Image, BAGEL, and Seedream 3.0, demonstrating the effectiveness of our method.

## 2 METHOD

In this section, we present details of our Plan-and-Paint framework. We begin by revisiting the prerequisite knowledge of Group Relative Policy Optimization (GRPO) algorithm in Sec. 2.1. Then, we introduce our Plan-and-Paint framework in Sec. 2.2 and Sec. 2.3, highlighting its core components: a prompt-level Chain-of-Thought (CoT) strategy and a noise-level reasoning methodology. In Sec. 2.4, we elaborate our multi-dimensional rewards design for effective reinforcemnet learning.

### 2.1 PRELIMINARY

Recently, reinforcement learning has emerged as a primary paradigm for enhancing the reasoning capabilities of large-scale models. Group Relative Policy Optimization (GRPO) (Guo et al., 2025) is a reinforcement learning algorithm designed to improve LLM reasoning by building upon Proximal Policy Optimization (PPO). Its primary contribution is a *group-relative* advantage estimation method that removes the need for a parameterized value function, thus enhancing training efficiency and stability. For each prompt, GRPO samples a group of responses from the current policy. The advantage $\hat{A}_i$ for each response is then computed by normalizing its scalar reward $r_i$ against the mean and standard deviation of the rewards within its peer group:

$$\hat{A}_i = \frac{r_i - \mu_{\mathcal{G}}}{\sigma_{\mathcal{G}}}, \quad \text{where} \quad \mu_{\mathcal{G}} = \frac{1}{G} \sum_{j=1}^{G} r_j, \quad \sigma_{\mathcal{G}} = \sqrt{\frac{1}{G} \sum_{j=1}^{G} (r_j - \mu_{\mathcal{G}})^2}. \tag{1}$$

The policy parameters $\theta$ are updated by maximizing the following objective function:

$$J_{\text{GRPO}}(\theta) = \mathbb{E}_{q \sim \mathcal{D}, \{o_i\}_{i=1}^{G} \sim \pi_{\theta_{\text{old}}}(\cdot|q)}$$
$$\left[ \frac{1}{G} \sum_{i=1}^{G} \min \left( \rho_i(\theta) \hat{A}_i, \text{clip}(\rho_i(\theta), 1 - \epsilon, 1 + \epsilon) \hat{A}_i \right) - \beta D_{\text{KL}}(\pi_\theta || \pi_{\text{ref}}) \right], \tag{2}$$

where $\rho_i(\theta) = \frac{\pi_\theta(o_i|q)}{\pi_{\theta_{\text{old}}}(o_i|q)}$ is the probability ratio for the entire sequence, $\epsilon$ is a clipping hyperparameter that constrains policy updates, and the KL divergence term $D_{\text{KL}}$ acts as a regularizer to prevent policy $\pi_\theta$ from deviating too far from a pre-trained reference model $\pi_{\text{ref}}$.

### 2.2 ADAPTIVE LENGTH PREDICTION FOR COT

To address the challenge of generating high-fidelity images from text prompts, we introduce a novel two-stage generation paradigm Plan-and-Paint, as illustrated in Fig. 2. This paradigm emulates the

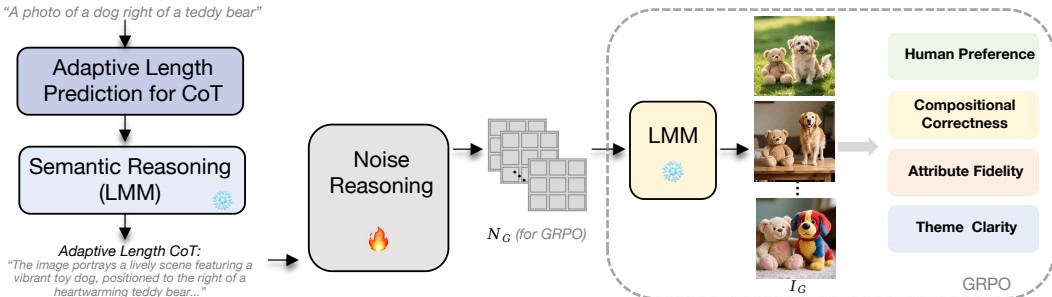

Figure 2: **Overview of Plan-and-Paint.** Given the input text prompt, the Adaptive Length Prediction module first executes the two-stage self-querying process to determine $L_{\text{opt}}$ to guide the semantic reasoning prompt CoT generation. This prompt CoT is then fed into the noise reasoning model to produce G initial noise maps $N_G$, using an autoregressive architecture. These noise maps are subsequently passed to a large multimodal model Qwen-Image to synthesize G corresponding images $I_G$. Finally, the generated images are evaluated by an ensemble of multi-dimensional vision experts to compute group-relative rewards and perform GRPO training.

human cognitive process of conceptualization before creation by first leveraging a MultiModal Large Language Model (MLLM) to generate a detailed Chain-of-Thought (CoT) narrative. This narrative serves as a rich, descriptive blueprint for the subsequent image synthesis stage.

A pivotal challenge in this approach is determining the optimal length of semantic CoT. A fixed-length strategy is suboptimal, as it fails to adapt to the widely varying semantic complexity of user prompts. An overly short CoT may omit critical details, while an excessively long one risks introducing contradictory information or semantic drift. To address this challenge, we propose Adaptive Length Prediction for Chain-of-Thought (ALP-CoT), an innovative mechanism that dynamically predicts the ideal reasoning length at inference time. Uniquely, ALP-CoT does not rely on an external, pre-trained regression model. Instead, it leverages the inherent reasoning capabilities of the MLLM itself through a structured, two-step self-querying process.

### 2.2.1 TWO-STAGE SELF-QUERYING MECHANISM

The core of ALP-CoT is a *SemanticLengthPredictor* module that instructs the MLLM to analyze its own task and prescribe a suitable reasoning budget. This process unfolds in two sequential stages:

**Stage 1: Semantic Task Classification.** The predictor first categorizes the user prompt $\mathcal{P}_{\text{user}}$ into predefined semantic types—e.g., *color*, *position*, *count*, *relation*, or *default*—by querying the MLLM with a structured prompt $\mathcal{Q}_{\text{classify}}$:

$$\mathcal{T}_{\text{task}} = \text{MLLM}\big(\mathcal{Q}_{\text{classify}}(\mathcal{P}_{\text{user}}), L_{\text{short}}\big), \tag{3}$$

where $\mathcal{T}_{\text{task}}$ denotes the identified task type, providing a strong contextual prior for the subsequent length prediction stage.

**Stage 2: Task-Specific Length Prediction and Calibration.** With the task type $\mathcal{T}_{\text{task}}$ identified, a second, more specific query, $\mathcal{Q}_{\text{predict}}$, is constructed. This query primes the MLLM to act as an expert for the given task type and recommend an optimal max reasoning length (in tokens) for the original prompt $\mathcal{P}_{\text{user}}$:

$$L_{\text{raw}} = \text{MLLM}\big(\mathcal{Q}_{\text{predict}}(\mathcal{P}_{\text{user}}, \mathcal{T}_{\text{task}}), L_{\text{medium}}\big). \tag{4}$$

To enhance stability and prevent erratic predictions, this raw value is calibrated using a set of predefined heuristics stored in *task_profiles*. Each task type $\mathcal{T}_{\text{task}}$ is associated with a base length $\beta_{\mathcal{T}}$ and a scaling factor $\sigma_{\mathcal{T}}$. The final optimal length $L_{\text{opt}}$ is computed as:

$$L_{\text{opt}} = \text{clip}\left(\lfloor \beta_{\mathcal{T}} + \sigma_{\mathcal{T}} \cdot L_{\text{raw}} \rfloor, L_{\text{min}}, L_{\text{max}}\right), \tag{5}$$

where clip$(\cdot)$ ensures $L_{\text{opt}}$ lies within $[L_{\text{min}}, L_{\text{max}}]$, balancing flexibility and stability. This calibration step grounds the MLLM's abstract recommendation in a well-defined numerical space, blending the model's dynamic reasoning with robust and rule-based constraints.

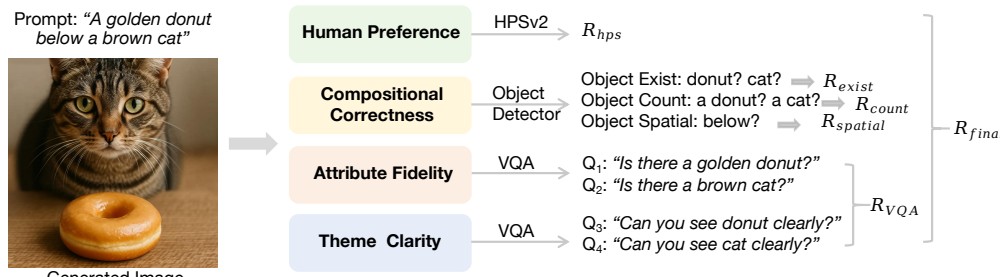

Figure 3: **Illustration of Rewards Design.** The diagram illustrates that our rewards assess the aesthetic quality, the semantic and compositional spatial fidelity to the prompt, as well as the image's theme alignment with the prompt.

#### 2.2.2 Integration into the Generation Pipeline

The ALP-CoT mechanism is seamlessly integrated as a precursor to the main CoT generation. At inference time, the *SemanticLengthPredictor* executes this two-stage self-querying process to determine $L_{\text{opt}}$. The main CoT reasoning is then performed with the *max_new_tokens* parameter explicitly set to this dynamically predicted value. This MLLM-driven, self-adaptive approach ensures that the reasoning depth is precisely tailored to the complexity of each prompt, significantly improving the robustness and quality of our Plan-and-Paint generation framework. We accompany an illustrative example of ALP-CoT in Appendix C.

### 2.3 Noise-level Reasoning

Beyond prompt-level reasoning, we introduce a reasoning paradigm that operates directly on the initial noise prior, which we term Noise-level Reasoning. In diffusion frameworks, particularly those employing the flow matching training objective like Qwen-Image (Wu et al., 2025a), the initial noise tensor $z \sim \mathcal{N}(0, I)$ is not merely a random starting point; it fundamentally dictates the global structure, composition, and key attributes of the final image. Motivated by this observation and inspired by NoiseAR (Li et al., 2025), we conceptualize the initial noise not as unstructured entropy, but as a latent canvas where the model's foundational decisions are encoded. This process is analogous to a sculptor selecting a block of marble, where its intrinsic properties profoundly influence the final sculpture. By applying GRPO optimization strategy within the initial noise space, we empower the model to perform reasoning at the most foundational level of generation. This allows it to learn an optimal noise prior that is already biased towards fulfilling the complex compositional requirements of the prompt, resulting in improvements in both prompt alignment and overall image fidelity.

### 2.4 Generation Rewards Design

Unlike rule-based reward mechanisms commonly used in language models, image evaluation cannot rely solely on predefined rules, as it requires a multifaceted assessment that includes aesthetic quality, object presence, semantic attributes, relational accuracy, and theme clarity. Given the complexity of such an evaluation, we employ an ensemble of vision-language experts to measure generated images from diverse perspectives. As shown in Fig. 3, our reward integrates the following components:

**Human Preference Metrics** ($R_{\text{HPS}}$). To ensure the holistic quality and prompt coherence of the generated images, we employ Human Preference Score v2 (HPSv2) model (Wu et al., 2023), which aims to align text-to-image synthesis with human preferences by predicting the likelihood of a synthesized image being preferred by users. We define this reward as $R_{\text{HPS}}(I_{gen}, P)$, which provides a crucial, high-level signal to guide our model toward producing visually compelling and contextually appropriate results.

**Compositional Correctness** ($R_{\text{Det}}$). Accurately generating compositional elements specified in a prompt—such as object existence, count, and spatial relationships—remains a primary challenge for text-to-image models. To address this, we employ the open-vocabulary object detector Ground-

ingDINO (Liu et al., 2024) as a specialized composition expert. For a prompt $P$ that specifies a set of $K$ target objects $\{o_i\}_{i=1}^{K}$, we formulate a composite reward signal, $R_{\text{Det}}$, as a weighted sum of multiple components. The foundational component is an existence reward $R_{\text{exist}}$:

$$R_{\text{exist}} = \frac{1}{K} \sum_{i=1}^{K} \mathbb{I}(\max(\text{conf}(o_i, I_{\text{gen}})) > \tau), \tag{6}$$

where $\text{conf}(o_i, I_{\text{gen}})$ yields the confidence scores of all detected instances of object $o_i$ in the generated image $I_{\text{gen}}$, $\tau$ is a predefined confidence threshold, and $\mathbb{I}(\cdot)$ is the indicator function.

When the prompt also dictates object counts or spatial relations (e.g., "three dogs to the left of a cat"), we introduce a count reward ($R_{\text{count}}$) and a spatial reward ($R_{\text{spatial}}$). $R_{\text{count}}$ measures the normalized difference between detected and requested object counts, while $R_{\text{spatial}}$ evaluates the geometric arrangement of bounding boxes (e.g., via relative coordinate checks). The total reward is then computed as $R_{\text{Det}} = w_1 R_{\text{exist}} + w_2 R_{\text{count}} + w_3 R_{\text{spatial}}$, providing a comprehensive and granular signal for structural fidelity.

**Attribute Fidelity and Theme Clarity** ($R_{\text{VQA}}$). Beyond structural correctness, fidelity to fine-grained attributes (e.g., color, texture) and theme clarity are crucial for generation quality. We employ a Visual Question Answering (VQA) model, GIT (Wang et al., 2022), as an attribute expert to assess this dimension. Instead of performing complex semantic parsing, we rephrase key descriptive phrases from the prompt $P$ into a set of $K$ verification questions. For example, a prompt containing "a black dog and a yellow cat" would yield the questions, $Q_1$: "Is there a black dog?", $Q_2$: "Is there a yellow cat?", $Q_3$: "Can you see dog clearly?" and $Q_4$: "Can you see cat clearly?". The VQA model then evaluates the generated image $I_{\text{gen}}$ against each question $Q_i$, providing a probability distribution over the answers "Yes" and "No". The final attribute fidelity reward aggregates the confidence in the affirmative answer across all questions:

$$R_{\text{VQA}} = \frac{1}{K} \sum_{i=1}^{K} P_{\text{VQA}}(\text{Yes}|I_{\text{gen}}, Q_i). \tag{7}$$

This approach effectively transforms the attribute verification and theme clarification task into a series of binary VQA problems, encouraging the model to correctly bind attributes to their corresponding objects.

**Final Reward Formulation.** The final reward $R_{\text{final}}$ for a given sample is a weighted average of the scores from these three expert models, creating a balanced and comprehensive training signal:

$$R_{\text{final}} = R_{\text{HPS}} + R_{\text{Det}} + R_{\text{VQA}}. \tag{8}$$

## 3 EXPERIMENT

### 3.1 EXPERIMENT SETUP

**Training Settings.** Our training dataset consists of text prompts sourced from T2I-R1 (Jiang et al., 2025), totaling 6,786 prompts with no images. We use the pre-trained semantic-level model in T2I-R1 as LMM to infer prompt-level CoT. Our base model is NoiseAR (Li et al., 2025), and we use Qwen-Image (Wu et al., 2025a) as image generator. In our GRPO training setup, we use a learning rate of 1e-6, and a beta of 0.01. For each input, we sample a group of N = 8 candidates.

**Benchmark.** We evaluate on GenEval (Ghosh et al., 2023) and WISE (Niu et al., 2025) benchmarks. GenEval contains 553 prompts across six compositional tasks (object generation, counting, color, spatial relations, attribute binding) for fine-grained text-to-image alignment evaluation. WISE includes 1,000 prompts requiring common sense reasoning in cultural concepts, spatial-temporal scenes, and natural science. We follow the official evaluation settings of all the benchmarks.

### 3.2 QUANTITATIVE EVALUATION

We present a comprehensive evaluation of our method against the vast majority of leading text-to-image models, spanning both **open-source and closed-source** projects across both **original and**

Table 1: Quantitative Evaluation Results on **GenEval**.

| Model | Single Object | Two Object | Counting | Colors | Position | Attribute Binding | Overall↑ |
|---|---|---|---|---|---|---|---|
| PixArt-$\alpha$ (Chen et al., 2024) | 0.98 | 0.50 | 0.44 | 0.80 | 0.08 | 0.07 | 0.48 |
| Emu3-Gen (Wang et al., 2024b) | 0.98 | 0.71 | 0.34 | 0.81 | 0.17 | 0.21 | 0.54 |
| TokenFlow-XL (Qu et al., 2025) | 0.95 | 0.60 | 0.41 | 0.81 | 0.16 | 0.24 | 0.55 |
| SDXL (Podell et al., 2023) | 0.98 | 0.74 | 0.39 | 0.85 | 0.15 | 0.23 | 0.55 |
| Janus (Wu et al., 2025b) | 0.97 | 0.68 | 0.30 | 0.84 | 0.46 | 0.42 | 0.61 |
| SD3-Medium (Esser et al., 2024) | 0.98 | 0.74 | 0.63 | 0.67 | 0.34 | 0.36 | 0.62 |
| FLUX (Wang et al., 2025) | 0.97 | 0.79 | 0.71 | 0.77 | 0.18 | 0.42 | 0.62 |
| JanusFlow (Ma et al., 2025) | 0.97 | 0.59 | 0.45 | 0.83 | 0.53 | 0.42 | 0.63 |
| GoT (Fang et al., 2025) | 0.99 | 0.69 | 0.67 | 0.85 | 0.34 | 0.27 | 0.64 |
| FLUX.1-dev (Labs, 2024) | 0.98 | 0.81 | 0.74 | 0.79 | 0.22 | 0.45 | 0.66 |
| DALL-E 3 (Betker et al., 2023) | 0.96 | 0.87 | 0.47 | 0.83 | 0.43 | 0.45 | 0.67 |
| Show-o (Xie et al., 2024) | 0.98 | 0.80 | 0.66 | 0.84 | 0.31 | 0.50 | 0.68 |
| FLUX+Pref-GRPO (Wang et al., 2025) | 0.99 | 0.86 | 0.74 | 0.81 | 0.26 | 0.57 | 0.70 |
| SD3.5 Large (Esser et al., 2024) | 0.98 | 0.89 | 0.73 | 0.83 | 0.34 | 0.47 | 0.71 |
| Show-o2-1.5B (Xie et al., 2025) | 0.99 | 0.86 | 0.55 | 0.86 | 0.46 | 0.63 | 0.73 |
| GoT-R1-7B (Duan et al., 2025) | 0.99 | 0.94 | 0.50 | 0.90 | 0.46 | 0.68 | 0.75 |
| T2I-R1 (Jiang et al., 2025) | 0.99 | 0.92 | 0.52 | 0.88 | 0.72 | 0.62 | 0.77 |
| Janus-Pro-7B (Chen et al., 2025) | 0.99 | 0.89 | 0.59 | 0.90 | 0.79 | 0.66 | 0.80 |
| BAGEL (Deng et al., 2025) | 0.99 | 0.94 | 0.81 | 0.88 | 0.64 | 0.63 | 0.82 |
| GPT-Image-1 [High] (OpenAI, 2025) | 0.99 | 0.92 | 0.85 | 0.92 | 0.75 | 0.61 | 0.84 |
| Seedream 3.0 (Gao et al., 2025) | 0.99 | 0.96 | **0.91** | **0.93** | 0.47 | 0.80 | 0.84 |
| Qwen-Image (Wu et al., 2025a) | 0.99 | 0.92 | 0.89 | 0.88 | 0.76 | 0.77 | 0.87 |
| Ours w/o ALP-CoT | 0.99 | 0.92 | 0.90 | 0.89 | 0.79 | **0.83** | 0.88 |
| Ours w/o NR | 0.99 | 0.96 | 0.84 | 0.90 | 0.80 | 0.81 | 0.88 |
| Ours | **1.00** | **0.98** | 0.90 | 0.91 | **0.82** | 0.77 | **0.90** |

Table 2: Quantitative Evaluation Results on **WISE**.

| Model | Cultural | Time | Space | Biology | Physics | Chemistry | Overall↑ |
|---|---|---|---|---|---|---|---|
| JanusFlow (Ma et al., 2025) | 0.13 | 0.26 | 0.28 | 0.20 | 0.19 | 0.11 | 0.18 |
| Janus (Wu et al., 2025b) | 0.16 | 0.26 | 0.35 | 0.28 | 0.30 | 0.14 | 0.23 |
| Show-o (Xie et al., 2024) | 0.28 | 0.40 | 0.48 | 0.30 | 0.46 | 0.30 | 0.35 |
| Janus-Pro-7B (Chen et al., 2025) | 0.30 | 0.37 | 0.49 | 0.36 | 0.42 | 0.26 | 0.35 |
| Emu3 (Wang et al., 2024b) | 0.34 | 0.45 | 0.48 | 0.41 | 0.45 | 0.27 | 0.39 |
| Harmon-1.5B (Wu et al., 2025c) | 0.38 | 0.48 | 0.52 | 0.37 | 0.44 | 0.29 | 0.41 |
| SDXL (Podell et al., 2023) | 0.43 | 0.48 | 0.47 | 0.44 | 0.45 | 0.27 | 0.43 |
| SD3-Medium (Esser et al., 2024) | 0.43 | 0.50 | 0.52 | 0.41 | 0.53 | 0.33 | 0.45 |
| SD3.5 Large (Esser et al., 2024) | 0.44 | 0.50 | 0.58 | 0.44 | 0.52 | 0.31 | 0.46 |
| PixArt-$\alpha$ (Chen et al., 2024) | 0.45 | 0.50 | 0.48 | 0.49 | 0.56 | 0.34 | 0.47 |
| Playground-v2.5 (AI, 2024) | 0.49 | 0.58 | 0.55 | 0.43 | 0.48 | 0.33 | 0.49 |
| FLUX.1-dev (Labs, 2024) | 0.48 | 0.58 | 0.62 | 0.42 | 0.51 | 0.35 | 0.50 |
| BAGEL (Deng et al., 2025) | 0.44 | 0.55 | 0.68 | 0.44 | 0.60 | 0.39 | 0.52 |
| T2I-R1 (Jiang et al., 2025) | 0.56 | 0.55 | 0.63 | 0.54 | 0.55 | 0.30 | 0.54 |
| Qwen-Image (Wu et al., 2025a) | 0.62 | **0.63** | 0.77 | **0.57** | **0.75** | **0.40** | 0.62 |
| Ours | **0.65** | 0.62 | **0.78** | 0.55 | 0.69 | **0.40** | **0.63** |

**RL** methods, on the GenEval and WISE benchmarks (in Table 1 and Table 2). Our method demonstrates substantial improvements over the baseline, achieving remarkable performance on GenEval (0.90) and on WISE (0.63), thereby establishing a new state-of-the-art. Notably, on the GenEval, our method secures **the top rank**, outperforming existing methods by a significant margin, including top-tier competitors such as GPT-Image-1 [High] (OpenAI, 2025), Janus-Pro-7B (Chen et al., 2025), Qwen-Image (Wu et al., 2025a), BAGEL (Deng et al., 2025), Seedream 3.0 (Gao et al., 2025), etc. What's more, on GenEval, our method leads in three of six subtasks, with an exceptional performance in the Position subtask (0.82) and Attribute Binding subtask (0.83), all surpassing previous SOTA results by over 3%, as shown in Table 1.

## 3.3 QUALITATIVE EVALUATION

Fig. 4 presents a comprehensive qualitative analysis comparing our method against baseline methods, including Qwen-Image, NoiseAR, T2I-R1, and our ablation study settings. We evaluate on

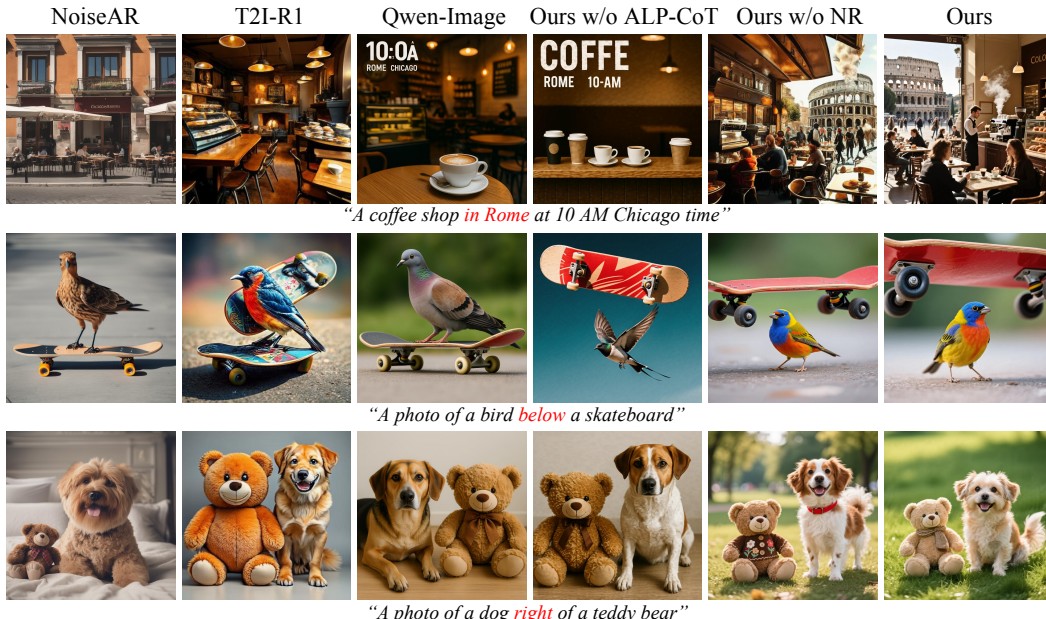

NoiseAR      T2I-R1      Qwen-Image      Ours w/o ALP-CoT      Ours w/o NR      Ours

*"A coffee shop in Rome at 10 AM Chicago time"*

*"A photo of a bird below a skateboard"*

*"A photo of a dog right of a teddy bear"*

Figure 4: **Visualization Results.** Qualitative comparison among the base model NoiseAR, T2I-R1, Qwen-Image, Ours w/o ALP-CoT, Ours w/o NR, and Ours full model. Our model demonstrates superior performance on prompt alignment and excellent image quality.

challenging prompts to test complex compositional reasoning, spatial relationships, and contextual understanding. More visualization results refer to Fig. 6 and Fig. 7 in the Appendix.

As shown in Fig. 4, for *"A coffee shop in Rome at 10 AM Chicago time"* (top row), baseline methods (NoiseAR, T2I-R1, Qwen-Image) generate generic cafés, failing to capture the location *"in Rome"*. In contrast, our approach and its variant without noise reasoning (NR) both generate scenes with recognizable Roman landmarks, demonstrating contextual understanding capabilities of our ALP-CoT. For *"A photo of a bird below a skateboard"* (second row), all baseline models incorrectly place the bird *on* the skateboard. Our method, and all its variants, correctly interprets this spatial relationship, demonstrating our effectiveness in addressing complex spatial composition.

We attribute these quantitative and qualitative improvements to two key innovations: our novel ALP-CoT mechanism, which improves context-aware instruction following ability, and significantly enhances output diversity as demonstrated in Fig. 5. And our advanced noise-level reasoning framework, which enhances the model's robustness and generative precision. Together, these contributions not only achieve a new state-of-the-art but also demonstrate a novel and effective pathway toward building more robust and precise text-to-image generation.

## 3.4 ABLATION STUDIES

We conduct systematic ablation studies to investigate key components of our approach in Table 3, and qualitative comparisons in Figure 4. We utilize Qwen-Image (Wu et al., 2025a) as our baseline, which achieves an overall score of 0.87, and establish a strong baseline for visual reasoning tasks.

**Direct Use Semantic CoT.** Directly applying Semantic-level CoT from T2I-R1 (Jiang et al., 2025) to Qwen-Image results in significant performance degradation, with the overall score dropping from 0.87 to 0.83. This 4% decrease demonstrates that naively transferring reasoning patterns across different model architectures introduces suboptimal reasoning chains.

**Effect of Noise Reasoning.** The integration of Noise Reasoning (NR) provides moderate improvement, with the overall performance from 0.83 to 0.84. NR achieves perfect scores in Single Object recognition (1.00) and notable gains in Colors understanding (0.93). However, substantial deficiencies persist in spatial reasoning (Position: 0.58) and compositional understanding (Attribute Binding: 0.72), indicating that while NR helps mitigate some transfer issues, it cannot fully address the fundamental limitations of fixed-length reasoning.

Table 3: Ablation Studies on GenEval.

| Model | Single Object | Two Object | Counting | Colors | Position | Attribute Binding | Overall↑ |
|---|---|---|---|---|---|---|---|
| Qwen-Image (Wu et al., 2025a) | 0.99 | 0.92 | 0.89 | 0.88 | 0.76 | 0.77 | 0.87 |
| Qwen-Image+Semantic CoT (Jiang et al., 2025) | 0.99 | 0.96 | 0.85 ↓ | 0.89 | 0.65 ↓ | 0.66 ↓ | 0.83 ↓ |
| Qwen-Image+NR+Semantic CoT | 1.00 ↑ | 0.97 ↑ | 0.86 ↓ | 0.93↑ | 0.58 ↓ | 0.72 ↓ | 0.84 ↓ |
| Qwen-Image+NR+Semantic CoT (L=30) | 0.98 ↓ | 0.91 ↓ | 0.81 ↓ | 0.85 ↓ | 0.73 ↓ | 0.76 ↓ | 0.84 ↓ |
| Qwen-Image+NR+Semantic CoT (L=77) | 0.98 ↓ | 0.94 ↑ | 0.74 ↓ | 0.93 ↑ | 0.69 ↓ | 0.80 ↓ | 0.84 ↓ |
| Qwen-Image+NR+Semantic CoT (L=1024) | 1.00 ↑ | 0.98 ↑ | 0.88 ↓ | 0.93 ↑ | 0.57 ↓ | 0.75 ↓ | 0.85 ↓ |
| Qwen-Image+NR+Semantic CoT (L=2048) | 1.00 ↑ | 0.98 ↑ | 0.88 ↓ | 0.93 ↑ | 0.57 ↓ | 0.75 ↓ | 0.85 ↓ |
| Qwen-Image+NR+ALP-CoT | 1.00 ↑ | 0.98 ↑ | 0.90 ↑ | 0.91 ↑ | 0.82 ↑ | 0.77 | 0.90 ↑ |

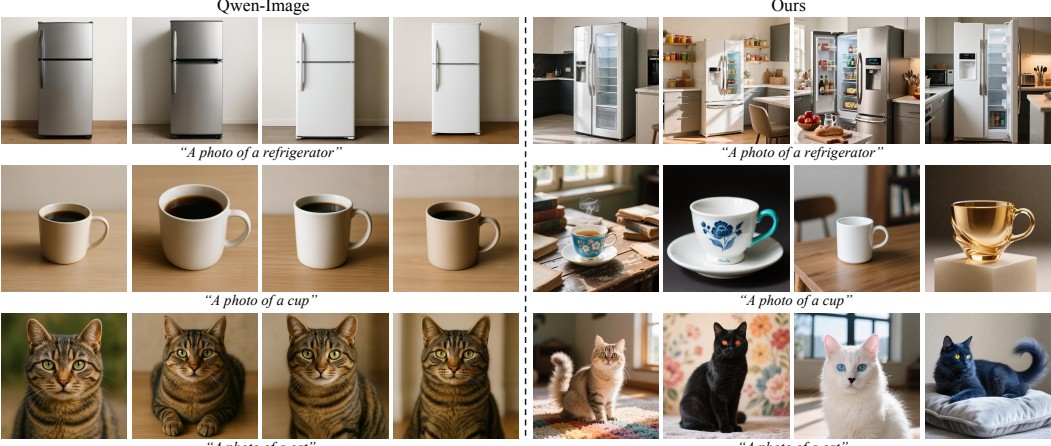

Figure 5: **Visualization Result of the Image Diversity of a Single Prompt.** We showcase the result of the baseline model Qwen-Image and our method.

**Analysis of Fixed-Length Constraints.** We systematically explore the effect of maximum token length $L$ (default=512) in Semantic-level CoT, varying $L$ from 30 to 2048 tokens. The results reveal a clear trade-off: shorter constraints ($L$=30) cause catastrophic failures across most metrics, while longer constraints ($L$=1024/2048) improve object-related tasks (Single Object: 1.00, Two Object: 0.98, Colors: 0.93) but severely harm spatial reasoning (Position: 0.57). This paradoxical behavior suggests that fixed-length reasoning fundamentally struggles to balance detailed object description with precise spatial and compositional understanding.

**Superiority of Adaptive Length Planning.** Our proposed ALP-CoT approach achieves superior performance (Overall: 0.90) by dynamically adapting reasoning length. It demonstrates significant improvements in the challenging Position task (0.82, **+17%** relative to fixed-length variants) while maintaining strong performance across all metrics. This improvement over the baseline validates that adaptive length planning is crucial for effective visual reasoning, particularly for tasks requiring complex spatial and compositional understanding.

## 4 CONCLUSION

In this work, we present Plan-and-Paint, a novel framework that sets a new state-of-the-art in text-to-image generation. Our method's strength lies in the synergy of two core components: an Adaptive Length Prediction for CoT (ALP-CoT) mechanism that tailors prompt complexity to enhance semantic alignment, and a Noise-level Reasoning process that ensures structural integrity. Trained using a GRPO framework with multi-dimensional rewards, our model achieves superior SOTA performance on the challenging GenEval and WISE benchmarks. Through extensive ablation studies, we confirm that ALP-CoT is crucial for semantic accuracy and Noise-level Reasoning for coherence. Together, they achieve a superior balance between prompt fidelity and image quality, providing a robust foundation for future research on reasoning-enhanced generative models.

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

## A    RELATED WORK

**Text-to-Image Generation.**    The field of text-to-image generation has witnessed remarkable progress through diffusion models (Saharia et al., 2022; Podell et al., 2023; Wang et al., 2025) and autoregressive approaches (Sun et al., 2024a; Li et al., 2024; Tian et al., 2024). While these models demonstrate impressive capabilities in generating high-quality images from text prompts, they exhibit significant limitations in compositional reasoning tasks (Fang et al., 2025). Complex prompts involving multiple objects with specific attributes and spatial relationships often lead to attribute binding errors, object omissions, and relationship violations. Recent efforts have attempted to address these issues through improved architectures (Fang et al., 2025; Duan et al., 2025) and training strategies (Jiang et al., 2025), yet the fundamental challenge of integrating structured reasoning into the generation process remains largely unsolved.

**Multimodal Large Language Models.** MLLMs (Achiam et al., 2023; Wang et al., 2024a; OpenAI, 2024) have made significant strides in bridging visual understanding and language processing. These models typically employ vision encoders (e.g., CLIP (Radford et al., 2021)) for visual feature extraction and large language models for reasoning and response generation. A growing research direction focuses on unifying visual understanding and generation within single models. Some approaches leverage external diffusion models for image synthesis (Sun et al., 2024b), while others utilize discrete tokenization methods (Esser et al., 2021) but face challenges in maintaining both generation quality and understanding capability. Dual-encoder architectures (Team, 2024) attempt to separate these tasks, yet effectively translating complex reasoning into high-quality visual generation remains an open challenge. Current methods primarily use MLLMs for prompt enhancement or preliminary planning (Deng et al., 2025), lacking deep integration of reasoning throughout the generation process.

**Reinforcement Learning for T2I Generation.** Reinforcement Learning has emerged as a powerful paradigm for enhancing reasoning capabilities in generative models. The success of reasoning-based RL approaches in language domains like OpenAI o1 (OpenAI, 2024) and DeepSeek-R1 (Guo et al., 2025) has inspired applications in multimodal settings. Group Relative Policy Optimization (GRPO) (Guo et al., 2025) provides an efficient framework for policy improvement through relative reward comparisons among candidate outputs, eliminating the need for separate critic networks. Recent work has begun exploring RL for compositional image generation (Duan et al., 2025; Jiang et al., 2025), employing rule-based rewards and multi-level optimization strategies. These approaches typically focus on either prompt-level reasoning or pixel-level refinement, but lack mechanisms for seamless coordination between high-level semantic reasoning and low-level noise reasoning. Our framework addresses this gap by introducing a unified reward ensemble that simultaneously optimizes semantic planning coherence and execution fidelity, enabling more effective translation of complex reasoning into high-quality visual outputs.

## B    MORE QUALITATIVE EVALUATIONS

We present more qualitative analysis in Fig. 6 and Fig. 7, which provides an extensive comparison of text-to-image generation capabilities across multiple strong methods, including NoiseAR (Li et al., 2025), T2I-R1 (Jiang et al., 2025), BAGEL (Deng et al., 2025), Flux-1-Konext-Pro (Labs, 2024), Qwen-Image (Wu et al., 2025a), and our approach. The evaluation spans five challenging prompts that test cultural understanding, object counting, spatial relationships, and compositional reasoning.

The first example, *"Traditional food for the Dragon Boat Festival in China"*, reveals significant limitations in cultural and contextual understanding among existing methods. While baseline models generate generic festival foods, only our approach correctly produces *zongzi* (rice dumplings), the traditional food specifically associated with this festival, demonstrating superior semantic reasoning ability in cultural knowledge representation.

In the second example, *"A photo of four computer keyboards"*, quantitative accuracy emerges as a key differentiator. All compared methods fail to generate exactly four keyboards, with most producing varying incorrect quantities. Our method alone achieves both precise numerical accuracy and high visual quality, highlighting our advantage in numerical reasoning and object counting tasks.

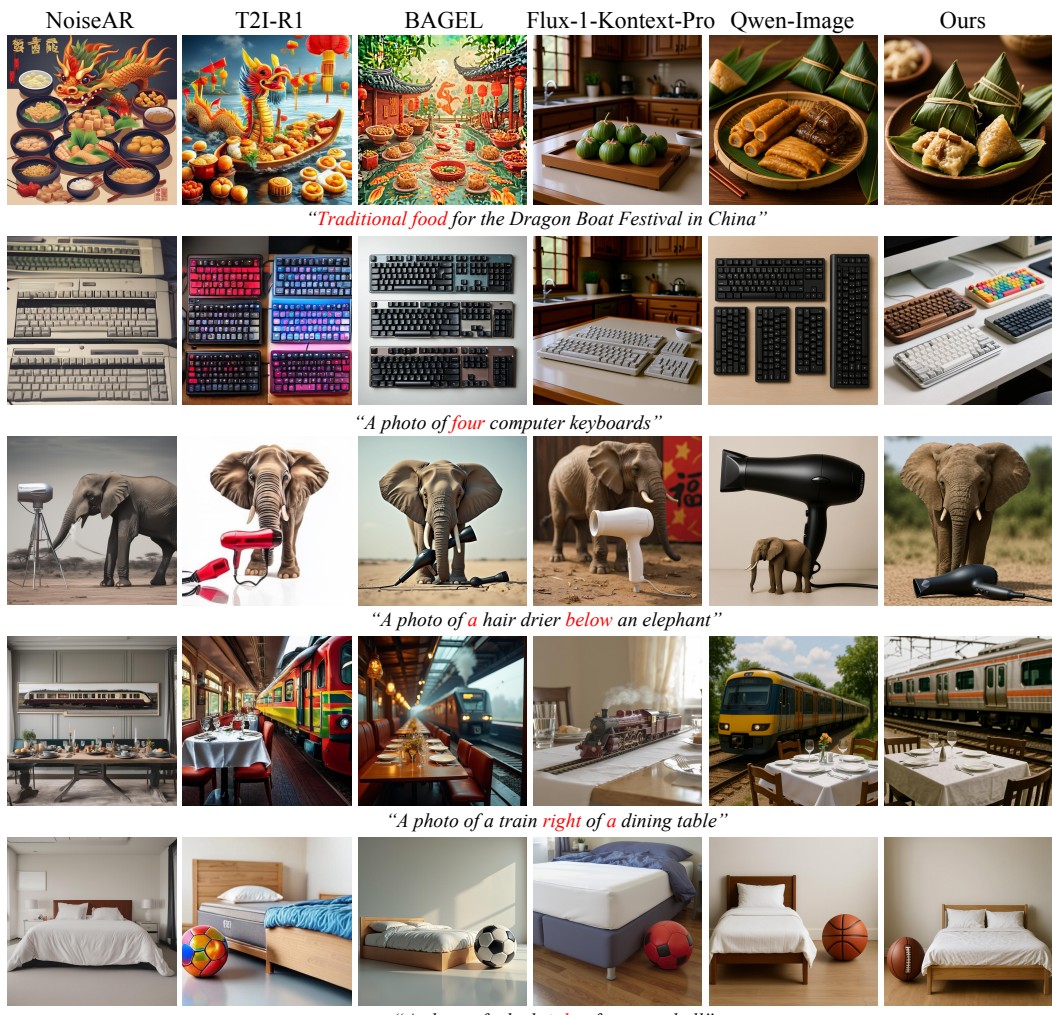

Figure 6: **Qualitative Comparisons.** Visual comparison of text-to-image generation results by NoiseAR, T2I-R1, BAGEL, Flux-1-Kontext-Pro, Qwen-Image, and our method. The results demonstrate our method's superiority in handling complex prompts involving cultural context (row 1: *Dragon Boat Festival food*), numerical accuracy (row 2: *four keyboards*), spatial relationships with object recognition (row 3: *hair drier below elephant*), and compositional reasoning (rows 4-5: *train right of dining table* and *bed right of sports ball*). Our approach consistently achieves accurate spatial relationships, right object counting, and high visual fidelity compared to baseline methods.

The third prompt, *"A photo of a hair drier below an elephant"*, presents a compound challenge requiring both spatial reasoning and object recognition. NoiseAR and Qwen-Image fail the spatial relationship, while T2I-R1, BAGEL, and Flux-1-Konext-Pro maintain correct spatial arrangement but generate incorrect objects instead of one hair dryer. Our approach uniquely satisfies both constraints—correct spatial positioning and accurate object representation.

The fourth example, *"A photo of a train right of a dining table"*, further emphasizes the spatial reasoning capabilities. While NoiseAR, Flux-1-Konext-Pro, and Qwen-Image produce incorrect spatial arrangements, and T2I-R1 achieves correct positioning but with poor image quality, our method generates both spatially accurate and visually coherent results, outperforming all alternatives.

The final prompt, *"A photo of a bed right of a sports ball"*, confirms our method's consistent superiority. Apart from T2I-R1 and our method, all other methods fail to interpret the spatial relationship correctly. Although T2I-R1 correctly interprets the spatial relationship, it fails to generate a recog-

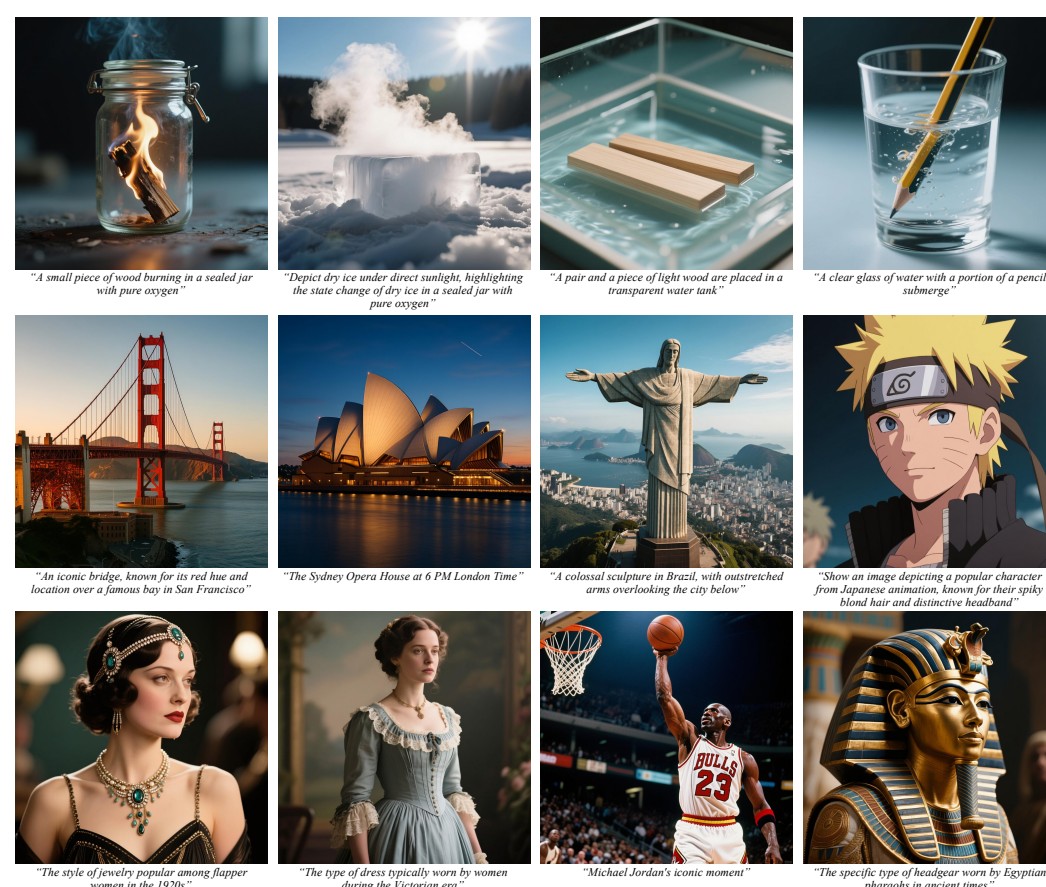

*"A small piece of wood burning in a sealed jar with pure oxygen"* | *"Depict dry ice under direct sunlight, highlighting the state change of dry ice in a sealed jar with pure oxygen"* | *"A pair and a piece of light wood are placed in a transparent water tank"* | *"A clear glass of water with a portion of a pencil submerge"*

*"An iconic bridge, known for its red hue and location over a famous bay in San Francisco"* | *"The Sydney Opera House at 6 PM London Time"* | *"A colossal sculpture in Brazil, with outstretched arms overlooking the city below"* | *"Show an image depicting a popular character from Japanese animation, known for their spiky blond hair and distinctive headband"*

*"The style of jewelry popular among flapper women in the 1920s"* | *"The type of dress typically worn by women during the Victorian era"* | *"Michael Jordan's iconic moment"* | *"The specific type of headgear worn by Egyptian pharaohs in ancient times"*

Figure 7: **Additional Qualitative Results.** Generated samples from Plan-and-Paint on diverse, complex prompts requiring multi-step reasoning, including physical processes (e.g., *"dry ice under direct sunlight"*), spatial compositions (*"light wood in a water tank"*), cultural and historic concepts (*"Flapper jewelry"*, *"Egyptian pharaoh headgear"*), and iconic scenes (*"Sydney Opera House at 6 PM London Time"*). These examples illustrate the model's capacity for structured and context-aware visual synthesis.

nizable bed. Our approach alone successfully satisfies both the spatial constraint and object fidelity requirements.

These comprehensive qualitative results demonstrate our method's absolute advantage across multiple dimensions of text-to-image generation, including cultural contextualization, numerical accuracy, spatial reasoning, object recognition, and overall visual quality. The consistent outperformance across diverse challenging prompts underscores the effectiveness of our proposed architectural innovations.

To further demonstrate the generalization capacity of our approach, we provide additional qualitative results in Fig. 7. As illustrated, Plan-and-Paint consistently generates coherent and contextually accurate images from a diverse set of challenging prompts. These include descriptions of complex physical phenomena (e.g., *"a small piece of wood burning in a sealed jar with pure oxygen"*), precise spatial arrangements (*"a pair and a piece of light wood placed in a transparent water tank"*), culturally rich concepts (*"the headgear of Egyptian pharaohs"*), and temporally-situated scenes (*"the Sydney Opera House at 6 PM London time"*). The model's ability to generate such a wide variety of concepts with high semantic alignment underscores the effectiveness of its dual-level reasoning mechanism—especially in decomposing abstract or composite instructions into structured generation plans. These results reinforce our claim that explicit hierarchical reasoning is essential to achieving robust and composable text-to-image generation.

## C    EXAMPLE OF ALP-COT

---

**Example: ALP-CoT in Action**

---

**Input User Prompt** ($\mathcal{P}_{\text{user}}$): *"A photo of a bird below a skateboard."*

**Stage 1: Semantic Task Classification**

- **Query** ($\mathcal{Q}_{\text{classify}}$): The MLLM is prompted to analyze the input and categorize its main challenge from a predefined list (e.g., *color*, *position*, *count*, *default*).
- **MLLM Analysis & Output** ($\mathcal{T}_{\text{task}}$): The model classifies the task as general-purpose.

```
default
```

**Stage 2: Task-Specific Length Prediction**

- **Query** ($\mathcal{Q}_{\text{predict}}$): Based on the "default" type, the MLLM is asked to predict a suitable reasoning length.
- **MLLM Predicted Length** ($L_{\text{opt}}$): The model returns the calibrated length for this task type.

```
64
```

**Result: Main CoT Generation**

- **Execution:** The main reasoning process is run with `max_new_tokens` set to 64.
- **Generated Semantic-CoT:**

```
A vibrant, colorful scene features a beautifully-detailed
black-and-white bird soaring gracefully below a sleek skateboard.  The
skateboard is positioned on a smooth, polished wooden surface, while
the bird adds a dynamic and intriguing touch to the composition.  The
skateboard's grip tape texture contrasts with the smooth feathers or
```

**Generated Image:**

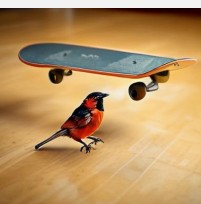

Figure 8: A step-by-step walkthrough of our Adaptive Length Prediction for CoT (ALP-CoT) mechanism. For the prompt *"A photo of a bird below a skateboard"*, the model predicts a "default" task type and allocates a 64-token budget, demonstrating its ability to tailor reasoning depth to the prompt's specific complexity.

**An Illustrative Example.** To make our ALP-CoT mechanism concrete, we walk through the process with the user prompt: *"A photo of a bird below a skateboard."* As detailed in Figure 8, the process begins with the *SemanticLengthPredictor* querying the MLLM to classify the prompt's primary semantic challenge. In this case, the MLLM categorizes the prompt as "default". This indicates that while a spatial relationship ("below") is present, the model does not deem it complex enough to require a specialist profile like "position", instead opting for a general-purpose reasoning budget. For the "default" task type, the MLLM predicts an optimal length of 64 tokens. This entire self-querying process occurs before the main CoT generation, ensuring the reasoning budget is tailored to the prompt's specific demands without being excessive. The MLLM then generates the final semantic CoT within this 64-token limit, which is subsequently used to guide the image synthesis stage.

**Baseline: Fixed-Length Semantic CoT.** For a direct comparison, we illustrate the process of the original T2I-R1 baseline model using the same prompt. As shown in Figure 9, it directly generates a Semantic CoT using a predefined, fixed token limit (e.g., 512 tokens) for all prompts, irrespective of their intrinsic complexity. This static approach often results in an overly detailed CoT, where the model invents extraneous attributes—a phenomenon we refer to as *reasoning hallucination*. For the prompt *"A photo of a bird below a skateboard"*, the model fabricates details like "a bold urban aesthetic adorned with stickers" and a "blurred, multicolored landscape", which can misguide the subsequent image generation process.

---

**Example: Baseline T2I-R1 in Action**

---

**Input User Prompt:** *"A photo of a bird below a skateboard."*

**Assistant: Semantic-CoT:**

```
The image portrays a vivid and dynamic scene showcasing a bird
gracefully perched beneath a skateboard, with various visual attributes
adding depth and detail to the composition.  The bird, characterized by
a rich palette of shades, exhibits a mix of vibrant blue, orange, and
white feathers with distinct textures, while the skateboard features
a bold urban aesthetic adorned with stickers and intricate patterns.
The skateboard, cast in a glossy metallic finish, displays multiple
colors and designs, including a prominent red stripe, and reflects the
surrounding light, creating a captivating interplay between textures and
hues.  The background is composed of a blurred, multicolored landscape,
evoking a sense of depth and motion, subtly emphasizing the vibrant
contrast between the stationary bird and the mobile skateboard.  This
creative juxtaposition brings a lively and engaging narrative to the
scene, while highlighting the harmonious coexistence between the natural
world and urban culture.
```

**Generated Image:**

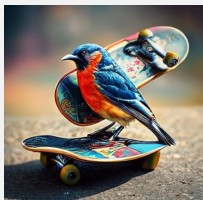

Figure 9: The generation process of the baseline T2I-R1 model. It uses a fixed-length semantic CoT mechanism. For a simple prompt, this leads to a verbose and hallucinatory CoT that includes unrequested details, contrasting sharply with the tailored output of our adaptive method.

## D    LIMITATIONS

While our method demonstrates strong performance, it inherits certain limitations common to large generative models. The planning module relies on the accuracy of prompt decomposition, which can occasionally fail on highly abstract or ambiguous instructions. Additionally, the training of our RL agent is computationally intensive, requiring significant resources that may hinder accessibility for some researchers. Future work could focus on optimizing the training efficiency to reduce computational costs while maintaining performance. Finally, our model's performance is bounded by the data it was trained on, and it may struggle with generating novel concepts or styles far outside its training distribution. Addressing these limitations presents valuable directions for future work.

## E  THE USE OF LARGE LANGUAGE MODELS

In accordance with ICLR policy, we disclose the use of Large Language Models (LLMs) in the preparation of this work.

- The LLM assisted solely in improving grammatical accuracy, sentence fluency, and academic tone. DeepSeek-V3.1 model (Guo et al., 2025) was used exclusively for language polishing and proofreading of early manuscript drafts.
- All scientific ideas, theoretical contributions, methodological designs, experimental results, and conclusions are entirely conceived and developed by the human authors. The LLM played no role in research ideation, technical innovation, or data analysis.
- We take full responsibility for the entire content of this manuscript. No LLM-generated content was used without thorough human review and editing.

## F  ETHICS STATEMENT

Our work presents a novel approach for text-to-image generation. While we only used publicly available datasets, we acknowledge that the capability of our model could potentially be misused for generating misleading content, such as deepfakes or copyrighted material without permission, if deployed irresponsibly.

To mitigate these risks, we commit to the following:

- The pre-trained models and code will be released strictly for research purposes under a license that prohibits malicious use.
- We strongly encourage the community to develop robust detection methods and attribution tools alongside generative technologies.

We believe the primary impact of our work is to advance the field of controllable content creation for positive applications like education, art, and design. We endorse the ongoing development of ethical guidelines for the safe deployment of generative AI.

## G  REPRODUCIBILITY STATEMENT

We have provided all details necessary to reproduce our results. Our models were trained on the dataset used in T2I-R1 (Jiang et al., 2025), and evaluated on GenEval (Ghosh et al., 2023) and WISE (Niu et al., 2025) benchmarks. The full model architecture and all critical hyperparameters (e.g., learning rates, batch sizes, reward function weights) are detailed in Sec. 2.2, Sec. 3.1 and Sec. 2.4. The training was conducted on $8 \times$ NVIDIA A6000 GPUs. We will release our source code, pre-trained model weights upon acceptance.

