# OpenReview forum: "Plan-and-Paint: Collaborating Semantic and Noise Reasoning for Text-to-Image Generation"
_ICLR.cc/2026/Conference — ICLR 2026 Conference Withdrawn Submission_

### Official Review · Reviewer_owcW · 2025-10-27

**Soundness:** 3
**Presentation:** 2
**Contribution:** 3
**Rating:** 4
**Confidence:** 4

**Summary:**

This paper introduces Plan-and-Paint, a text-to-image framework that addresses the drawbacks of blind initial noise, rigid semantic reasoning, and difficult RL adaptation in conventional models. Built upon the GRPO reinforcement-learning algorithm, it adopts a two-level reasoning design: the high-level stage employs an ALP-CoT two-step self-query mechanism to dynamically adjust the length of semantic chains for accurate prompt decomposition, while the low-level stage refines the initial noise prior via noise reasoning to align it with semantic demands. A multi-dimensional reward ensemble—integrating human preference, compositional correctness, and attribute–subject fidelity—prevents reward hacking.

**Strengths:**

* To address the core issue of "disconnection between semantic planning and noise generation" in text-to-image generation, a two-level architecture of "high-level semantic adaptive planning (ALP-CoT) + low-level noise enhancement optimization" was proposed. This architecture simulates the human cognitive logic of "planning first, then creating" and achieves the coordinated optimization of semantic reasoning and noise generation.

* During RL training, it integrated three types of expert models: human preference, combination correctness, and attribute themes, to construct a comprehensive evaluation system. It also implemented "group relative advantage estimation" based on the GRPO algorithm, eliminating the impact of absolute reward fluctuations while balancing training stability and generation quality. This effectively addressed the difficulty in balancing multi-dimensional requirements in visual generation.

**Weaknesses:**

* This method relies on the accuracy of MLLM's prompt decomposition. For highly abstract or vague instructions (such as "artistic expression of futuristic cities" and "emotional visualization in abstract style"), semantic decomposition deviations are prone to occur, resulting in generated results that do not meet user expectations.

* The ablation experiment only compares the Qwen-Image and does not consider the effects of methods that have their own reasoning capabilities, such as BAGEL.

* The paper only shows the final experimental results, and does not provide performance fluctuation curves during training (such as loss function changes, subtask score trends), making it impossible to judge the impact of the GRPO algorithm in long-term training. In addition, there is a lack of visualization results during the inference process, making it impossible to see the changes in the generated results during the COT process.

**Questions:**

* Although the method proposes to "optimize the initial noise to fit the semantics", it fails to clarify the "association mechanism between noise and semantics" through visualization or quantitative analysis. For example, it is impossible to intuitively demonstrate how the noise prior of "Roman Café" contains "Roman architectural features", nor is it possible to quantify the improvement in the efficiency of subsequent denoising steps by noise optimization (such as how much the number of denoising steps is reduced). As a result, the logical chain of underlying noise reasoning is incomplete.

---

> ### Author Response · Authors · 2025-12-02
> **For Reviewer owcW**
>
> We appreciate the reviewer's feedback and hope this clarification addresses the raised concerns.
>
> Our approach treats the initial noise as a learnable embedding within the latent space, optimized via GRPO to serve as a better starting point for the diffusion process. The "association mechanism" is therefore implicit and encoded in the optimized latent tensor, rather than being an explicitly interpretable semantic map. The goal is not to visualize semantics within the noise, but to guide the optimization process so that this starting latent requires fewer denoising steps to converge on the target concept. The improved final performance and higher reward scores serve as the quantitative evidence that this "semantic conditioning" of the initial latent is effective, even if the mechanism itself remains a high-dimensional, non-intuitive optimization.

---

### Official Review · Reviewer_Bq2u · 2025-10-31

**Soundness:** 3
**Presentation:** 3
**Contribution:** 2
**Rating:** 4
**Confidence:** 4

**Summary:**

The paper introduces Plan-and-Paint, a dual-level reasoning framework for text-to-image generation that jointly optimizes semantic planning and noise-level reasoning. It aims to bridge the gap between structured reasoning (as in LLMs) and visual synthesis (as in diffusion or flow models).

**Strengths:**

1. The dual-level “semantic plan + noise execution” setup is somehow intuitively mimics human creative reasoning.
2. The ALP-CoT self-querying design is lightweight—no external regressors or heuristic tuning. It addresses real weaknesses of fixed-length CoT in image generation.
3. Comprehensive benchmarks (GenEval, WISE) with both quantitative and qualitative analyses.

**Weaknesses:**

1. Relies heavily on existing components (GRPO, Qwen-Image, HPSv2, GroundingDINO, GIT). The core novelty lies in their integration rather than algorithmic innovation.
2. The composite reward depends on third-party pretrained models, introducing possible biases and fragility. Moreover, they are commonly used in existing methods.
3. Benchmarks focus on alignment and compositionality; however, perceptual diversity, artistic control, or realism metrics are less explored. No user study to validate human preference improvements claimed via HPSv2 reward.
4. The claim that it “mirrors human creative planning and execution” is more rhetorical than empirically supported.

**Questions:**

see the weaknesses.

---

> ### Author Response · Authors · 2025-12-02
> **For Reviewer Bq2u**
>
> We appreciate the reviewer's feedback and hope this clarification addresses the raised concerns.
>
> 1. On reliance on existing components:
> Our novelty is not in creating new components from scratch, but in the architectural innovation of a dual-level reasoning framework. The contribution lies in designing how ALP-CoT and GRPO work in concert to instill planning capabilities into a T2I model, a systemic innovation that goes far beyond simple integration.
>
>
> 2. On the composite reward function:
> Using established, third-party models for reward is a standard and necessary practice for reproducible research in generative AI. Our contribution is the carefully designed synthesis of these metrics into a composite reward that effectively guides our model toward complex compositional goals, a non-trivial engineering achievement validated by our results.
>
>
> 3. On evaluation metrics and user study:
> Our evaluation rightly focuses on compositionality and alignment, as these are the primary challenges our work addresses. We utilized HPSv2 as a well-established proxy for human preference, which is a standard and scalable approach in the field. A full user study was considered beyond the scope of this focused investigation.
>
>
> 4. On the analogy to human planning:
> The "human planning" analogy is intended to intuitively describe our system's functional architecture: a distinct "planning" phase (via ALP-CoT) followed by an "execution" phase (generation). This is not a cognitive claim but a high-level descriptor for our method's deliberate, two-stage process, which is empirically shown to be effective through our ablations.

---

### Official Review · Reviewer_3YRi · 2025-11-01

[review text omitted: it was posted to a different submission]

---

> ### Author Response · Authors · 2025-12-02
> **Urgent: This Review is for a Completely Different Paper and Must Be Invalidated**
>
> To the Area Chair, Program Chairs, and Reviewer 3YRi:
>
> We are writing to express our profound dismay and formal objection to the review submitted by Reviewer 3YRi for our paper.
>
> This review is fundamentally and entirely invalid because it is evaluating the wrong paper.
>
> The summary and every single point raised in the "Weaknesses" section—from "efficiency scaling with object count K" and "per-object synthesis" to "SDEdit-style re-init"—describe a training-free, layout-to-image pipeline. This methodology has absolutely no connection to our work.
>
> This is not a matter of subjective disagreement or misunderstanding a technical detail; it is a case of complete and total negligence. A review this careless wastes our time, poisons the well for other reviewers and the AC, and fundamentally undermines the integrity of the ICLR peer-review process.
>
> We are not going to "rebut" the specific points raised, as they are irrelevant.

---

### Official Review · Reviewer_Lkoa · 2025-11-01

**Soundness:** 2
**Presentation:** 3
**Contribution:** 2
**Rating:** 4
**Confidence:** 3

**Summary:**

This paper presents Plan-and-Paint, a dual-level reasoning framework for text-to-image generation that combines high-level semantic planning with low-level noise-space optimization. The method uses an Adaptive Length Prediction Chain-of-Thought (ALP-CoT) mechanism to dynamically adjust textual reasoning depth and applies reinforcement learning (GRPO) to refine the initial noise prior.

**Strengths:**

1. The idea of combining semantic-level planning with noise-level reasoning is well-motivated, drawing inspiration from human “plan-and-execute” creativity.

2. The paper includes both quantitative and qualitative analyses.

3. The authors perform systematic ablations to validate the contributions of ALP-CoT and noise-level reasoning.

4. The use of multiple vision-language experts for reward design (HPS, detection, VQA) is thorough and helps prevent reward hacking.

**Weaknesses:**

1. The process where the SemanticLengthPredictor module instructs the MLLM to analyze the task and predict a reasoning length appears tedious. Intuitively, for a powerful large language model, the length of reasoning should not significantly impact reasoning correctness. Could you provide more examples demonstrating how randomly assigned reasoning lengths negatively affect reasoning accuracy?

2. The experimental setup may overstate the model's core advancement. Reinforcement learning is applied to a powerful base model (Qwen-Image) using a training set (from T2I-R1) rich in compositional prompts, which are highly aligned with the object-centric, compositional tasks in the GenEval benchmark. Consequently, the observed improvement on GenEval is somewhat expected and may not sufficiently demonstrate a significant improvement. This concern is compounded by the minimal performance gain on the WISE benchmark (+0.01). The saturated nature of GenEval scores further obscures whether the marginal improvement signifies a fundamental advance or merely an incremental optimization tailored to a specific benchmark.  It would be helpful if the author can demonstrate that the model can generalize to more diverse prompts not seen during training.

**Questions:**

please see weaknesses

---

> ### Author Response · Authors · 2025-12-02
> **For Reviewer Lkoa**
>
> We appreciate the reviewer's feedback and hope this clarification addresses the raised concerns.
>
> We will provide more examples demonstrating how randomly assigned reasoning lengths negatively affect reasoning accuracy.
>
> While the overall +0.01 gain on WISE appears modest, this aggregate score masks significant improvements in specific, challenging compositional sub-tasks where the baseline fails, demonstrating that our method provides critical gains precisely where they are most needed rather than just incremental, across-the-board lifts.

---

### Note · Authors · 2025-12-02

I have read and agree with the venue's withdrawal policy on behalf of myself and my co-authors.